# Numerical Analysis of Multi-Angle Precision Microcutting of a Single-Crystal Copper Surface Based on Molecular Dynamics

**DOI:** 10.3390/mi13020263

**Published:** 2022-02-06

**Authors:** Jianhe Liu, Liguang Dong, Junye Li, Kun Dong, Tiancheng Wang, Zhenguo Zhao

**Affiliations:** 1Ministry of Education Key Laboratory for Cross-Scale Micro and Nano Manufacturing, Changchun University of Science and Technology, Changchun 130022, China; a_liu@cust.edu.cn (J.L.); dlg147@126.com (L.D.); dkdk2022@163.com (K.D.); wtc202201@163.com (T.W.); zjg202201@163.com (Z.Z.); 2Chongqing Research Institute, Changchun University of Science and Technology, Chongqing 401135, China

**Keywords:** molecular dynamics, abrasive flow precision machining, microcutting, cutting angle, cutting mechanism

## Abstract

The molecular dynamics method was used to study the removal mechanism of boron nitride particles by multi-angle microcutting of single-crystal copper from the microscopic point of view. The mechanical properties and energy conversion characteristics of single-crystal copper during microcutting were analyzed and the atomic displacement and dislocation formation in the microcutting process are discussed. The research results showed that during the energy transfer between atoms during the microcutting process of boron nitride particles, the crystal lattice of the single-crystal copper atom in the cutting extrusion region was deformed and displaced, the atomic temperature and thermal motion in the contact area between boron nitride particles and Newtonian layer of workpiece increased, the single-crystal copper atom lattice was defective, and the atomic arrangement structure was destroyed and recombined. The interface of different crystal structures formed a dislocation structure and produced plastic deformation. With the increase of the impact cutting angle, the dislocation density inside the crystal increased, the defect structure increased and the surface quality of the workpiece decreased. To protect the internal structure of the workpiece and improve the material removal rate, a smaller cutting angle should be selected for the abrasive flow microcutting function, which can reduce the formation of an internal defect structure and effectively improve the quality of abrasive flow precision machining. The research conclusions can provide a theoretical basis and technical support for the development of precision abrasive flow processing technology.

## 1. Introduction

As an approach to nontraditional finishing technology for improving surface quality, abrasive flow precision machining technology plays an important role in the field of precision and ultra-precision machining technology. Research on abrasive flow precision machining technology is also increasing. Han Sangil and other scholars conducted abrasive flow processing on the internal channel surface of 15–5 ph stainless steel with abrasive particles of different sizes and concentrations, revealing their possible wear mechanism and movement [1]. Sushil Mittal and other scholars designed the optimal parameter combination for the abrasive flow processing of aluminum-based SiC metal matrix composites [2]. Zhang L and other scholars discussed the study of the constrained abrasive flow precision machining of complex titanium alloys by numerical and experimental results, and found that the triangular restraint plate can greatly improve the efficiency and quality of precision machining [3]. Duval Chaneac and other scholars designed and fabricated clamp tubes and selective laser melting (SLM) components that can be disassembled and assembled during abrasive flow processing, and studied the surface roughness evolution on the inner surface of SLM components [4]. Gov Kursad and other scholars used four types of abrasive media to prepare four groups of abrasive media and studied the effect of abrasive type on the abrasive flow process [5]. Sambharia JK and other scholars have proposed a novel one-way abrasive flow processing setup and another abrasive flow processing medium for finishing industrial parts, which can better remove pits and cracks on the surface of the workpiece [6]. Dong ZG and other scholars studied the processing mechanism of high-viscosity elastic abrasive flow processing and derived the removal rate of abrasive flow processing materials with high viscoelastic grinding media [7]. Wang TT and other scholars have developed a new AFM device. The numerical simulations have studied the effects of the dynamic viscosity of the abrasive medium, the gap distance and the speed of the tool bar on the flow field pressure and flow velocity distribution. The influence of an abrasive medium and key parameters on the finishing effect of the new AFM was investigated experimentally [8]. Wei HB and other scholars proposed a new atomic force microscope material removal prediction model, which can accurately predict the change trend of the profile height change ΔH and mass change AM of material removal [9].

Regarding the research on abrasive flow precision machining technology, most of the research work is from a macroscopic point of view; there is less research on abrasive flow precision machining technology from the micro point of view. As a theoretical research method at the micro level, molecular dynamics simulation is an effective experimental tool for studying nanoscale particle cutting. In order to explore the intrinsic microscopic surface creation mechanism of the precision surface of abrasive flow, domestic and foreign scholars have done relevant research. Yuan S and other scholars used molecular dynamics simulation to study the atomic removal mechanism of chemical mechanical polishing of diamond powder on a diamond surface in an H_2_O_2_ aqueous solution [10]. Dai HF and other scholars used molecular dynamics simulation to study a new single-crystal silicon mechanical polishing method using laser-made nanostructured diamond abrasives. Compared with traditional mechanical polishing, the changes of polishing force, atomic displacement and stress during nanopolishing were studied [11]. Oren and other scholars conducted a molecular dynamics simulation of the screw dislocations of the heart-cubic copper crystal under certain stresses and temperatures and determined a new cross-slip mechanism [12]. Zhou P and other scholars used molecular dynamics simulation to study the removal mechanism of SiC during fixed abrasive polishing. It was found that the deeper the depth of grinding, the deeper the surface damage [13]. Li Y and other scholars used molecular dynamics simulation to clarify the formation mechanism of residual stress and surface roughness of an ultra-precision diamond-cut single crystal crucible [14]. Alhafez IA and other scholars systematically studied the single-crystal iron nanocutting process, and found that the edge dislocations formed at the cutting edge slide obliquely to it, resulting in a complex three-dimensional dislocation network [15]. Yue XM and other scholars simulated the single discharge process of polycrystalline copper with molecular dynamics. The formation and evolution process of the defect structure were analyzed [16]. Zhang JJ and other scholars simulated the ultraprecision diamond cutting process by molecular dynamics, and explained the potential mechanism of cerium in ultraprecision diamond cutting [17]. Fang QH and other scholars used molecular dynamics to simulate the nanomachining processing of Cu/Ag double-layer and pure Cu film and analyzed the influence of processing parameters on subsurface damage and material removal [18]. Dai HF and other scholars conducted a comprehensive comparison of traditional nanoscale machining (TM) to laser-assisted nanoscale machining (LAM) by using molecular dynamics simulation. LAM can obtain a higher quality workpiece surface and is a technology with potential [19]. Dong GJ and other scholars used molecular dynamics to simulate the diamond-mirror cutting of large-diameter aluminum alloy, revealing the wear mechanism in the cutting process [20]. Zou L and other scholars used molecular dynamics to study the catalytic simulation of metal iron on the transformation of diamond crystals into a graphite structure and analyzed the main causes of diamond tool wear [21].

In the process of abrasive flow precision machining, the abrasive particles suspended in the medium randomly impact the surface of the workpiece at a certain speed with the flow of the medium. The schematic diagram of the abrasive microcutting workpiece is shown in Figure 1. In previous studies, most of them chose diamond as the abrasive particle, considering the high cost of diamond. Therefore, the molecular dynamics method was used to study the removal mechanism of boron nitride particles by multi-angle microcutting of single-crystal copper from the microscopic point of view. It can provide a theoretical basis and technical support for the development of precision abrasive flow processing technology. The mechanical properties, energy conversion, atomic displacement and dislocation formation of abrasive particles at different cutting angles were analyzed. The removal law of surface materials in the microcutting process was revealed. The influence of different cutting angles on the internal structure of the workpiece is discussed. The polishing mechanism of abrasive flow at micro scale was revealed.

## 2. Model Establishment and Selection of Potential Function

### 2.1. Establishment of Microcutting Model for Boron Nitride Particles

Constructing a molecular dynamics simulation model of abrasive microcutting was the first step in numerical simulation of molecular dynamics. In order to explore the influence of different angles on the microcutting process, boron nitride was selected as the abrasive grain, and single-crystal copper was used as the workpiece material to construct a simulation model of boron nitride particles grain microcutting of the single-crystal copper workpiece material. As shown in Figure 2, the simulation model of single-crystal copper was cut by two different angles of boron nitride particles grains. The model size was 144 Å × 163 Å × 87 Å, the abrasive radius was 15 Å, and the total atomic number in the model was 161058. Among them, the number of boron atoms was 1205 and the number of nitrogen atoms was 1220. The workpiece was divided into a fixed boundary layer, a constant temperature layer and a Newton layer, and periodic boundary conditions were adopted in the X direction of the workpiece to eliminate the boundary effect caused by the scale limitation of the simulation system. Before the simulation began, the simulation model was energy minimized to eliminate the unreasonable factors in the initial stage of modeling. The initial temperature of the analog system was set at 310 K. In order to balance the analog ensemble, the number of relaxation steps was set at 10,000 steps. The boron nitride particles grains were initially placed on the upper right side of the workpiece, the cutting speed of the abrasive grains was set to 80 m/s, the initial cutting depth was 15 Å, and the cutting angle was 5°, and the abrasive grains were cut in the -Y direction. The numerical simulation step of molecular dynamics was 100,000 steps, and the integration step size was Δt = 1 fs.

### 2.2. Selection of Potential Function

In this paper, EAM potential function is used to describe the interaction between copper atoms [22], Morse potential function is used to describe the interaction between copper atoms and boron nitride [23], Tersoff potential function is used to describe the interaction between boron nitride and copper atoms [24].

#### 2.2.1. EAM Potential Function

EAM potential is a typical multi-body potential suitable for a metal atomic system. It can describe the interaction between copper atoms more accurately. It has high accuracy in describing the structure and thermodynamic properties of metal. Therefore, EAM potential is widely used in molecular dynamics simulation of a metal atomic system. Its form of expression is as follows:(1)$$E=\frac{1}{2}\sum_{i,j}\phi_{ij}(r_{ij})+\sum_{i}F_{i}(\rho_{i})$$
(2)$$\rho_{i}=\sum_{j\neq i}\rho_{j}(r_{ij})$$

In the formula, $\phi_{ij}$ is the interaction potential between atoms, *r_ij_* is the distance between atoms *i* and *j*, $F_{i}$ is the embedded energy function of atoms *i*, $\rho_{i}$ is the sum of the electron densities of all atoms except atoms *i*, and *ρ_j_* is the electron density function of atoms *j* at *i*.

#### 2.2.2. Morse Potential Function

Morse potential is a typical pair potential model with simple analysis and good approximation to the fine structure of atomic vibration. Its expression is as follows:(3)$$u(r_{ij})=D[e^{-2\alpha (r_{ij}-r_{0})}-2e^{-\alpha (r_{ij}-r_{0})}]$$

In the formula, D is the binding energy, α is the elastic coefficient, *r_ij_* is the instantaneous atomic distance, and *r_0_* is the equilibrium atomic distance.

#### 2.2.3. Tersoff Potential Function

Tersoff potential function can reasonably describe the interaction between boron nitride atoms in the form of:(4)$$E=\sum_{i}E_{i}=\frac{1}{2}\sum_{i\neq j}V_{ij}$$
(5)$$V_{ij}=f_{C}(r_{ij})[f_{R}(r_{ij})+b_{ij}f_{A}(r_{ij})]$$
(6)$$f_{C}(r_{ij})=\{\begin{array}{c} 1,r_{ij}<R_{ij}\\ \frac{1}{2}+\frac{1}{2}\cos[\pi \frac{r_{ij}-R_{ij}}{S_{ij}-R_{ij}}],R_{ij}<r_{ij}<S_{ij}\\ 0,r_{ij}>S_{ij} \end{array}$$
(7)$$f_{R}(r_{ij})=A_{ij}e^{\left(-\lambda_{ij}r_{ij}\right)}$$
(8)$$f_{A}(r_{ij})=-B_{ij}e^{\left(-\mu_{ij}r_{ij}\right)}$$
(9)$$b_{ij}={\left(1+\beta_{i}^{n}\zeta_{ij}^{n}\right)}^{-1/2n}$$
(10)$$\zeta_{ij}=\sum_{k\neq i,j}f_{C}(r_{ik})g(\theta_{ijk})$$
(11)$$g(\theta_{ijk})=\frac{1+c^{2}/d^{2}-c^{2}}{\left[d^{2}+{\left(h-\cos\theta_{ijk}\right)}^{2}\right]}$$

In the formula, $E_{i}$ is the potential energy of atom *i*, $V_{ij}$ is the bond energy of atom *i* and *j*, *r_ij_* is the distance of atom *i* and *j*, *f_c_* is the truncation function, *f_R_* is the pair of exclusive potential, *f_A_* is the pair of attractive potential, *b_ij_* is the coefficient of attraction, *R* and *S* are the truncation length, $A_{ij}$ is the dual binding energy of exclusive term, and $\lambda_{ij}$ is the gradient coefficient of the dual potential curve of the exclusive term. For the dual binding energy of the attraction term, $B_{ij}$ is the gradient coefficient of the dual potential curve of the attraction term, $\mu_{ij}$ is the bond order coefficient, $\beta_{i}$ is the angular potential energy, $\zeta_{ij}$ is the bond angle between the bond *r_ij_* and *r_jk_*, and *c*, *d* and *h* are the elastic constants.

## 3. Analysis and Discussion

### 3.1. Cutting Force Analysis

In the process of boron nitride particles grains colliding with microcutting single- crystal copper workpiece materials, the boron nitride particle grains and the single crystal copper workpiece material interacted. In order to study the change law of abrasive cutting force during microcutting, the change of cutting force at different cutting angles was analyzed. The cutting force variation curve in each direction is shown in Figure 3.

The direction of [$\bar{1}$00] the cutting force is shown in Figure 3a. Since the cutting direction was [0$\bar{1}\bar{1}$], the cutting force in the [$\bar{1}$00] direction was friction, so it goes back and forth and up and down around 0 eV/Å. As the cutting distance increased, the cutting force fluctuated reciprocally and showed an increasing trend. This phenomenon is closely related to the degree of deformation of the crystal lattice, lattice reconstruction, and the generation of an amorphous phase transition. The direction of [0$\bar{1}$0] and [00$\bar{1}$] cutting force is shown in Figure 3b,c. The cutting force in the [00$\bar{1}$] and [0$\bar{1}$0] directions was the shear force. As the number of simulation steps increased, the cutting distance gradually increased, and the cutting force in the direction of [00$\bar{1}$] and the direction of [0$\bar{1}$0] fluctuated and gradually increased. After increasing to a certain value, it no longer increased, but was still in a wave state. In the initial stage of microcutting, atomic repulsive forces were generated between the abrasive particles and the workpiece atoms, which exhibited a small initial cutting force. To achieve the purpose of cutting, a larger cutting force was required to break the interaction between the workpiece atoms. After the abrasive particles were in contact with the workpiece, the cutting force required by the workpiece was gradually increased until the critical value of the bonding force between the atoms was exceeded. The lattice of the copper atoms was destroyed, and the chemically healthy fracture caused the copper atoms to move relative to each other. As the depth of cutting increased, the number of atomic interactions that need to be destroyed increased, and the cutting force tended to increase until the abrasive grains were completely cut into the workpiece, and the fluctuation state of the cutting force tended to be stable.

The cutting force in the direction of [0$\bar{1}$0] and the cutting angle of the abrasive grains were not in a positive linear relationship. The cutting force at the cutting angles of 0°, 5° and 10° was smaller than the other cutting angles, and the fluctuation amplitude was also smaller than the other cutting angles. This is because during the microcutting process with cutting angles of 0°, 5°, and 10°, the cutting depth of the abrasive grains on the whole workpiece material was smaller than other cutting angles, and the burr height of the workpiece material was between 3.5 Å and 15 Å. The cutting at a small angle was basically the removal of the burr, the cutting depth of the workpiece material itself was shallow, and the degree of damage of the crystal structure and the degree of deformation were relatively small, so the shear force in the direction of [0$\bar{1}$0] was small throughout the cutting process. The cutting force in the direction of [00$\bar{1}$] was different from the cutting force in the direction of [0$\bar{1}$0]. In the later stage of numerical simulation, the cutting force in the direction of [00$\bar{1}$] was stable, and the cutting angle and the cutting force were positively correlated. When the cutting angle was 0°, the cutting force was the smallest. As the collision cutting angle increased, the cutting force also increased. The reason is that the speed of the abrasive grains was 80 m/s; the cutting angle became larger and the sub-speed of the direction of [00$\bar{1}$] became larger. The larger the angle of the same simulation step, the larger the cutting depth and the more the number of atomic arrays destroyed.

### 3.2. Energy Analysis

In order to study the energy variation law of workpiece in the microcutting process, the potential energy, kinetic energy and total energy of boron nitride particles during microcutting single crystal copper were analyzed.

In order to study the influence of different cutting angles on the kinetic energy, the kinetic energy change in the simulation process was analyzed, and the work done by the abrasive particles to the workpiece in the microcutting process could be intuitively understood. Figure 4 shows the atomic kinetic energy curves at different cutting angles. The force between the workpiece atom and the abrasive particle was a long-range repulsive force during microcutting. As the distance between the abrasive particles and the workpiece atoms gradually decreased, the repulsion between the atoms gradually increased, and the copper atoms in the workpiece started to move to generate kinetic energy. When the number of simulation steps reached 38,000 steps, the kinetic energy of the atom fluctuated around 1609 ev. At this point the abrasive particles did not completely enter the workpiece. When the number of simulated steps reached 45,000 steps, the kinetic energy increased to 1678 ev. As the number of analog steps increased, the kinetic energy fluctuated around 1678 ev. When the abrasive particles were in contact with the surface of the workpiece, the atomic temperature of the contact zone increased, the lattice deformed, the coordinates changed, the displacement occurred, the thermal motion was enhanced, and kinetic energy was generated. As the abrasive particles began to enter the workpiece, the copper atoms in the workpiece began to move violently and the atomic kinetic energy increased. As the simulation progressed, the atomic strain energy of the workpiece entered a stable state, and the kinetic energy of the workpiece atom fluctuated up and down around a certain value, and the generation and transformation reached a dynamic equilibrium state. The cutting force drove the atomic temperature of the contact zone to increase, the thermal motion of the atom increased, and the kinetic energy also increased, resulting in a local increase in the kinetic energy of the copper atom. 

In order to study the effect of different cutting angles on the potential energy, the potential energy change in the simulation process was analyzed. Figure 5 shows the variation of the atomic potential energy of the workpiece at different cutting angles. The potential energy of copper atoms showed an increasing trend. In the initial stage of microcutting, the boron nitride particles grains moved forward. At this time, the repulsive force between the workpiece atoms and the abrasive particles was the initial cutting force, the workpiece atoms were compressed under the action of the repulsive force, and the strain energy accumulated in the crystal lattice was continuously increased. The workpiece atom displaced, the lattice and the crystal lattice was destroyed. When the strain energy increased to a certain value, the chemical bond between the workpiece atoms was destroyed, the original lattice structure was disrupted, the strain energy was released, and the workpiece atoms were forced to rearrange in the form of a low lattice, dislocation motion occurred in the workpiece, and strain energy was converted into potential energy. When the cutting angle was varied between 0°and 20°, the trend of potential energy was very obvious for the same number of simulation steps. When the cutting angle was varied between 25°and 45°, the potential energy curve was substantially closed throughout the process. The potential energy curve at the cutting angle of 0° was at the bottom. Since the change of potential energy and the generation of dislocations were related to the atomic motion, when the cutting angle angle was 0°, the dislocations generated by the cutting were the smallest. Therefore, a smaller cutting angle can reduce the formation of defect structures inside the crystal during microcutting.

In order to study the effect of different cutting angles on total energy, the change of total energy in the simulation process was analyzed. Figure 6 shows the variation of the atomic total energy of the workpiece at different cutting angles. The total energy of the simulated system was the sum of atomic kinetic energy and atomic potential energy. The kinetic energy change of the workpiece atom was related to the thermal motion of the atom. The change of the cutting angle had little effect on the thermal motion of the atom, and the kinetic energy did not change with the change of the cutting angle. Therefore, the trend of total energy was similar to the trend of atomic potential energy. Unlike the change of kinetic energy, the change of potential energy and total energy was obviously related to the cutting angle. The cutting angle increased, the potential energy of the workpiece atom increased, and the total energy also increased. 

### 3.3. Atomic Displacement Analysis

To inquire into the workpiece atomic displacement of abrasive cutting angle change, the influence of different cutting angles of the workpiece materials, grinding grain under function forms, the material removal process, the generation of chip and the change process, workpiece atoms according to atomic displacement coloring, the workpiece cutting area under different abrasive cutting angle profiled of atomic displacement, and local atomic displacement vector marking were analyzed, as shown in Figure 7.

In the microcutting process, the direction of atomic displacement was closely related to the cutting angle of the abrasive particles. When the cutting angle was small, the direction of atomic displacement basically followed the microcutting direction. As can be seen from the figure, during the microcutting process, the grinding particles perfored a microcutting action on the workpiece along the cutting direction, the workpiece atoms were squeezed by the grinding particles, the workpiece materials produced plastic deformation, the workpiece atoms produced displacement, gradually accumulated in the front end of the grinding particles, and finally removed from the surface of the workpiece in the form of chips. By comparing the atomic displacement of the workpiece at different cutting angles, it was found that with the increase of the cutting angle, the region producing the atomic displacement diffused from the surface of the workpiece to the interior of the workpiece, and the lattice deformation gradually increased. When the cutting angle was larger than 20°, a phenomenon in which some atoms move in the opposite direction in the cutting direction occurred, and this phenomenon was more prominent when the cutting depth was increased. When the abrasive grains cut the burrs on the surface of the workpiece, the FCC lattice structure in the area where the burrs were in contact with the surface of the workpiece was more broken; that is, when only the burr was removed, the cutting of the abrasive grains led to more breaking of the workpiece. The surface generated a certain lattice defect structure; the generation of atomic displacement led to the destruction and reorganization of the atomic arrangement structure, and when the adjacent atoms were displaced in a larger direction, the resulting structure was more complicated. The adjacent regions of the abrasive grains included an atomic arrangement structure, such as HCP and BCC. When a group of atoms moved in the same direction, the arrangement structure was an HCP structure, and the contact faces of different crystal structures formed a dislocation structure.

### 3.4. Dislocation Line Analysis

Some of the atoms in the crystal were subjected to external forces, and the sliding edges along a certain crystal plane and crystal orientation were formed according to a certain rule, thereby forming a dislocation structure. The generation of dislocations was closely related to the strength of the surface of the workpiece. During the low-speed microcutting process, the displacement of the workpiece atoms caused the destruction and reconstruction of the crystal lattice, causing dislocations and plastic deformation inside the workpiece. In order to study the lattice structure deformation mechanism inside the surface of the workpiece during the cutting process of boron nitride particles grains, a single boron nitride particles grain was selected as the object on the YOZ surface to generate different dislocation lines inside the workpiece during microcutting. And the change and the lattice structure were analyzed. Figure 7 displays the analysis of the bond angle and dislocation line of the single crystal copper during the microcutting process.

As shown in Figure 8, the simulation time was selected as 60 ps and 70 ps. According to the Ackland–Jones analysis method (Ackland–Jones analysis), atomic markers were labeled according to different arrangement structures of atoms to analyze the atomic displacement of the workpiece. Since the crystal lattice structure of single crystal copper was FCC, the atoms of FCC structure in the workpiece were removed to facilitate the analysis of lattice changes. In the figure, red, blue and light green represent HCP structure, BCC structure and amorphous structure, respectively. In the microcutting process, a large number of dislocation lines and lattice deformations occurred in the workpiece under the action of cutting forces, and more lattice transformed into HCP structures. The number of atoms of these structures increased with the increase of the cutting angle. During this phase transition process, the workpiece’s atomic strain kept increasing. When the atomic stress state exceeded the critical value of thermodynamic phase transition and was in a metastable state, the HCP phase began to form a nucleus and grew spontaneously, and the FCC lattice of copper became completely unstable, resulting in a sudden change in the mechanical quantity. With the progress of microcutting, due to the cutting and collision of grinding particles, the bonds between copper atoms broke, breaking the original lattice structure, and part of the copper atoms became disordered. At this time, this part of the atoms formed an amorphous structure. When the cutting angle increased from 0°to 45°, the cutting depth of the abrasive particles on the workpiece surface increased gradually, and the number of displaced atoms in the cutting area increased, and the original arrangement of more atoms changed, resulting in more lattice phase transitions. From the above analysis of energy, it can be seen that different cutting angles caused different changes in atomic energy, which indirectly affected the accumulation and release of strain energy in the material lattice, resulting in differences in the types and numbers of lattice phase transitions.

The dislocation extraction algorithm was used to identify different types of dislocation. Red arrows, green, light blue, pink and blue represented Burgers Vectors, 1/6<112>Shockley dislocation, 1/3<111>Frank dislocation, 1/6<110>Stair-Rod dislocation and 1/2<110>Perfect dislocation, respectively. During microcutting, due to the shear action of the abrasive particles, the workpiece atoms produced displacement, the original arrangement changed, and the local atoms misarranged, resulting in dislocation in the workpiece. It can be seen from the figure that most of the dislocations in the workpiece were mixed dislocations. The dislocation line changed, moved and multiplied around the cutting motion of the abrasive particles. The closer the dislocation line was to the position of the abrasive particles, the higher the density of the dislocation line was. Dislocation slipped along its Burgers Vectors, causing plastic deformation of the workpiece and changing the original configuration state and lattice phase transition. As microcutting went on, the number and types of dislocations in the workpiece material changed in a way such that dislocations proliferated. When the simulation time reached 60 ps, there were more 1/6<112>Shockley dislocations. When the simulation time reached 70 ps, the longer Shockley dislocations became less numerous, while the shorter Shockley dislocations became more numerous, and some of the original shorter linear dislocations became curved dislocations. This may be due to the gradual decomposition of dislocation proliferation. With the increase of the cutting angle, the number and density of dislocations increased obviously, and most of them were concentrated in the region where the lattice phase transition occurred. With the increase of the cutting angle, the dislocation density increased, and the defect structure inside the crystal also increased. The material was more prone to plastic deformation, and it was difficult to guarantee the machining quality of the workpiece. Therefore, reducing the cutting angle can improve the machining quality.

In order to more vividly describe the relationship between the cutting angle and the dislocation, the total length of the dislocation line generated during the cutting process of the boron nitride particles grains at different angles was analyzed, and the total length of the dislocation line generated during the microcutting process is shown in Figure 9.

It can be seen from Figure 9 that as the number of simulation steps increased, the number of dislocations increased, the total length became larger, and the cutting angle was positively correlated with the length of the total dislocation line. As the cutting angle increased, the total length of the dislocation lines also increased significantly. When the cutting angle was 0° and 5°, the length of the dislocation line was the shortest, and the number of dislocations generated was small. Therefore, a smaller cutting angle was advantageous for reducing the defect structure inside the workpiece, thereby obtaining a better cutting effect.

## 4. Conclusions

The mechanism of abrasive flow multi-angle precise microcutting of a single-crystal copper surface was discussed by a molecular dynamics method. A model of the microcutting action of boron nitride particles was established. The cutting force characteristics, atomic energy conversion, formation and variation of atomic displacement and dislocation of boron nitride particles in microcutting process are analyzed, and the multi-angle was revealed. The mechanism of multi-angle precision microcutting single crystal copper surface was revealed. The conclusions are as follows:

During the microcutting process, the cutting force generated by the abrasive particles fluctuates up and down, overcoming the bonding force between the atoms of the workpiece, destroying the atomic lattice of the workpiece so that the atomic bond is broken, and the atoms of the workpiece are displaced to achieve the purpose of material removal.

With the increase of the cutting angle, the cutting force gradually increases, and the abrasive particles have more difficulty cutting the workpiece. With the increase of the cutting angle, the atomic energy changes greatly, the atomic displacement changes become more complicated, and the displacement area becomes larger, making it more difficult to remove the chip from the workpiece surface. With the increase of the cutting angle, the number of dislocations and dislocation density increase.

When boron nitride particles cut a single-crystal copper workpiece at a cutting angle of 0° to 15°, the cutting force in [0$\bar{1}$0] direction and in [00$\bar{1}$] direction is relatively small, the cutting process is more flexible and stable, lattice phase transition is less, and dislocation number and density are less. A smaller cutting angle can reduce the formation of internal defect structures and improve the surface quality of the workpiece.

## Figures and Tables

**Figure 1 micromachines-13-00263-f001:**
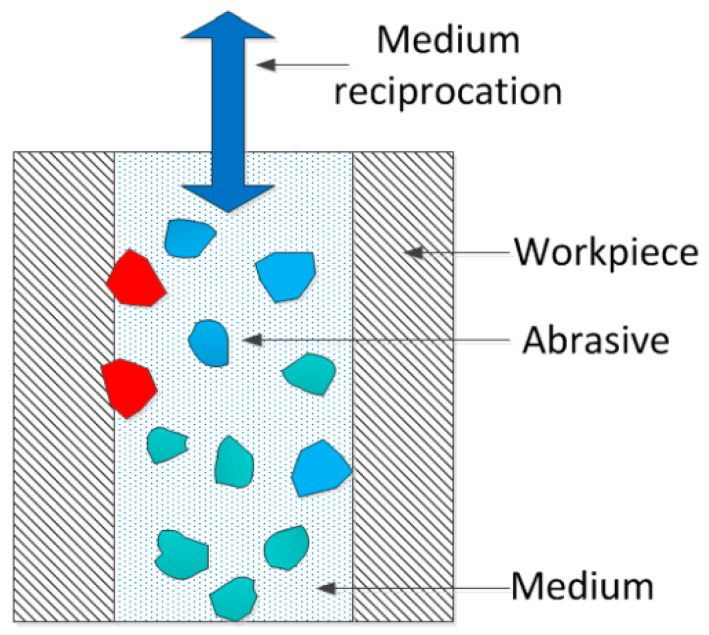
Abrasive grain microcutting workpiece schematic.

**Figure 2 micromachines-13-00263-f002:**
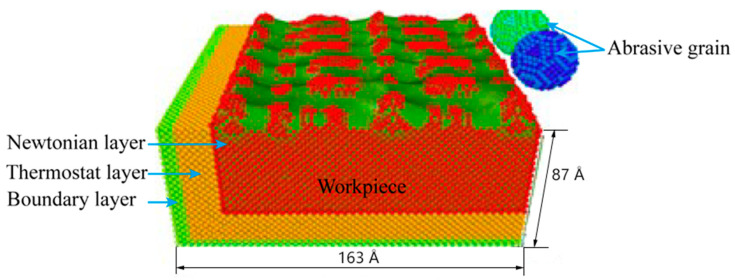
Numerical analysis model of boron nitride particles cutting.

**Figure 3 micromachines-13-00263-f003:**
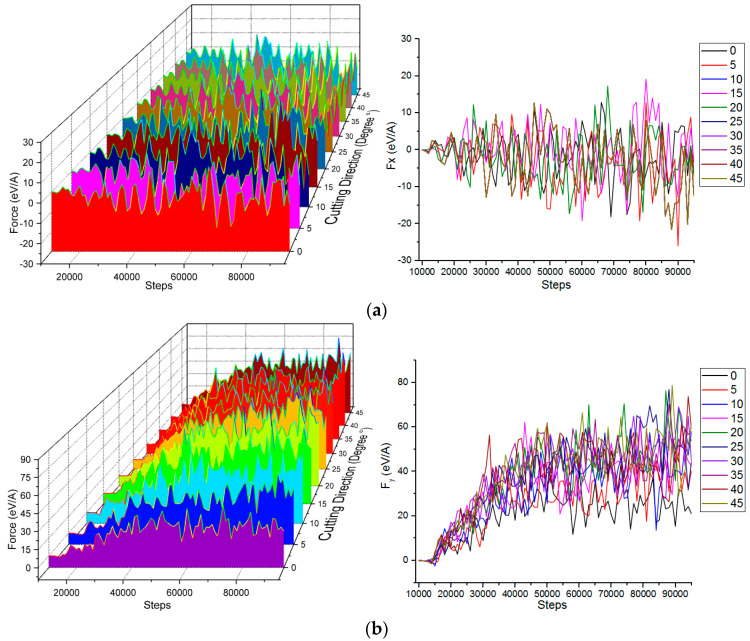
Cutting forces in different directions under various microcutting angles. (**a**) Cutting force in direction of [$\bar{1}$00]. (**b**) Cutting force in direction of [0$\bar{1}$0]. (**c**) Cutting force in direction of [00$\bar{1}$].

**Figure 4 micromachines-13-00263-f004:**
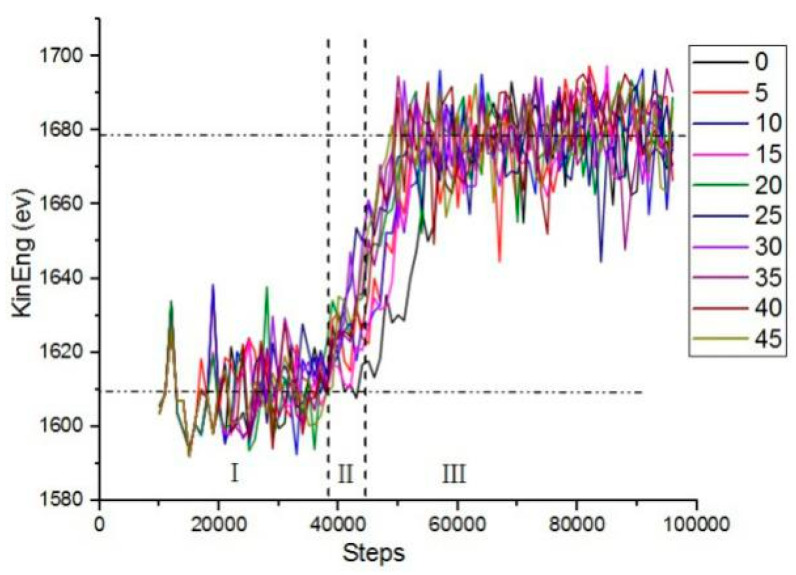
The variation of the atomic kinetic energy of the workpiece at different cutting angles.

**Figure 5 micromachines-13-00263-f005:**
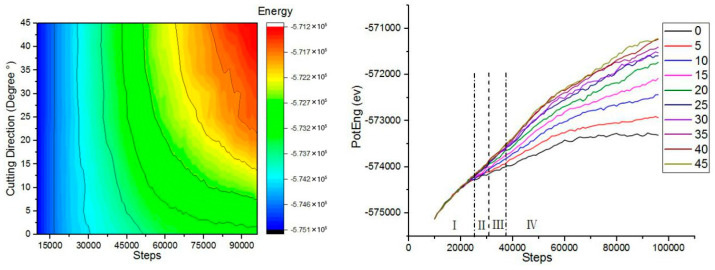
The variation of the atomic potential energy of the workpiece at different cutting angles.

**Figure 6 micromachines-13-00263-f006:**
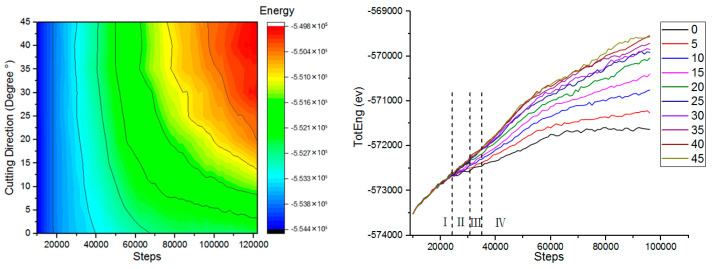
The variation of the atomic total energy of the workpiece at different cutting angles.

**Figure 7 micromachines-13-00263-f007:**
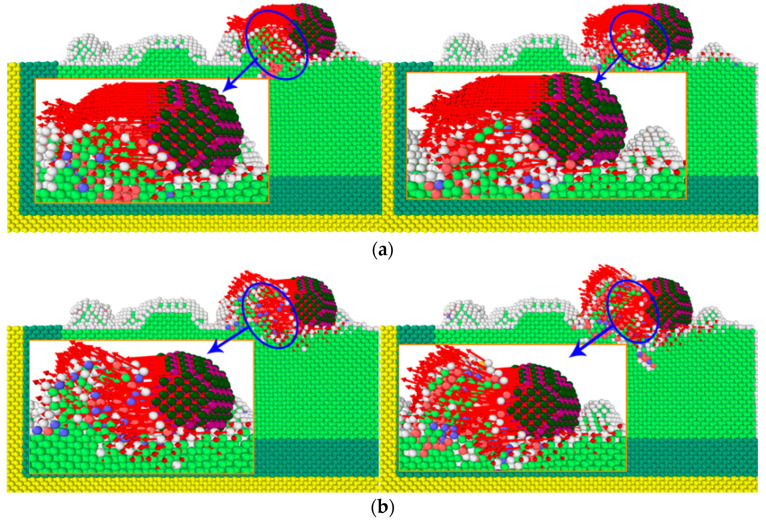
The atomic displacement of the workpiece at different cutting angles. Atomic colors: 
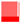
 Hexagonal Close-Packed (HCP) structure. 
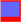
 Body-Centered Cubic (BCC) structure. 
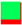
 Face-Centered Cubic (FCC) structure White: Amorphous structure. (**a**) 0°. (**b**) 5°. (**c**) 10°. (**d**) 15°. (**e**) 20°. (**f**) 25°. (**g**) 30°. (**h**) 35°. (**i**) 40°. (**j**) 45°.

**Figure 8 micromachines-13-00263-f008:**
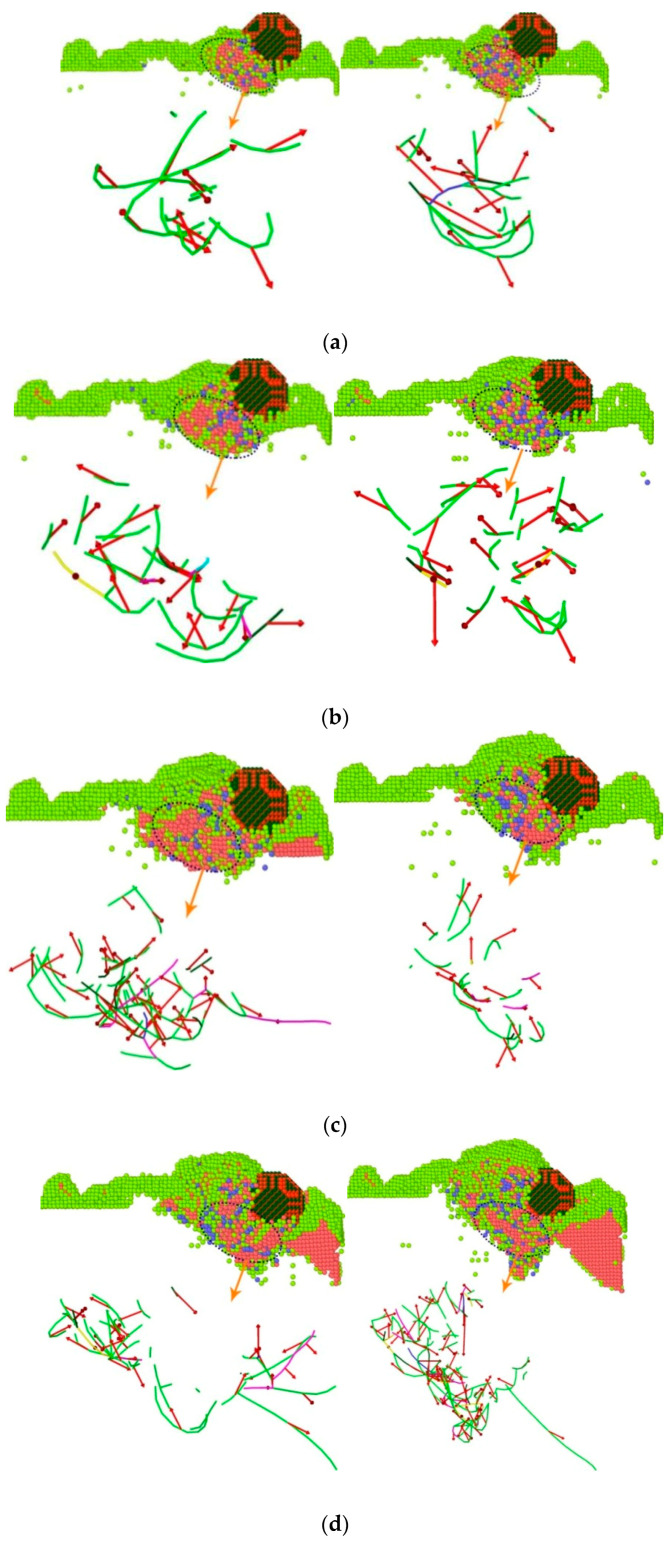
Atomic structure diagram and dislocation line of copper atoms with different microcutting angles. (**a**) 0°. (**b**) 5°. (**c**) 10°. (**d**) 15°. (**e**) 20°(**f**) 25°. (**g**) 30°. (**h**) 35°. (**i**) 40°. (**j**) 45°.

**Figure 9 micromachines-13-00263-f009:**
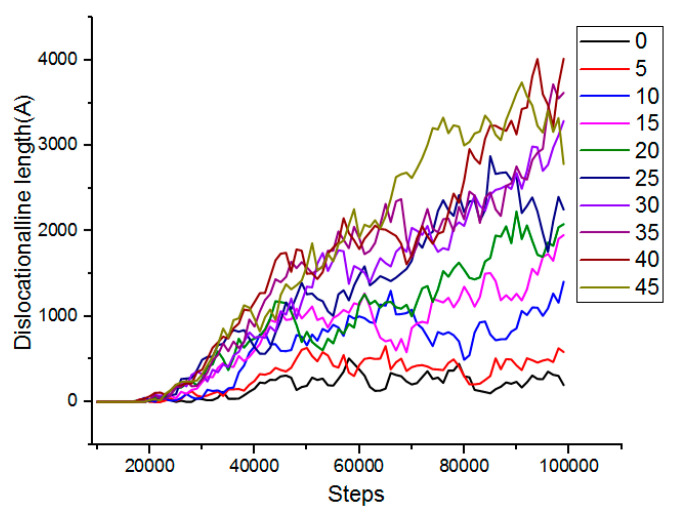
Total length of dislocation lines generated during micro-machining.

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
