# Peer review of "Numerical Analysis of Multi-Angle Precision Microcutting of a Single-Crystal Copper Surface Based on Molecular Dynamics"

_micromachines, 2022, doi:10.3390/mi13020263_

Round 1

Reviewer 1 Report

From the reading of the work it emerges that it is a modeling of the AFM process.
If so, several questions arise:
1) Why are monocrystalline copper and boron nitride abrasive particles used as work materials?
2) Why is it stated that the model is for these materials and not for others?
3) What is the particle size used in practice and how does the model represent that size?
4) Are the authors sure that the speed of the process is 80 m/s?
5) How is it possible in practice to vary the angle of impact of the particles?

Reviewer 2 Report

Dear Authors, 

I have some comments on your work:

  1. Please describe the novelty of your paper more in detail.
  2. What was the applied software to perform the simulation?
  3. Please label all of the components in the Figure 2.
  4. Line140: What does the "EAM" stand for?
  5. Line 179 and 180: It should be Fig. 3(b) and Fig. 3(c). Please correct it.
  6. Line 209: It is hard to understand the next phrase: "The reason is when the abrasive grain The speed is 80m/s.". Please check it.
  7. Lines 241,242 and 248 and 249 and abstract: The authors claimed about the changes in atomic temperature and thermal motion. How did they measure the temperature? Please provide some data (in the form of table or graphs, .. ) to show these changes during simulation process.
  8. In Fig 7. what do the "HCP", 'BBC" and "FCC" stand for?
  9. Nothing is mentioned about the atomic temperature and thermal motion in the conclusion section.

Best regards

Reviewer

Round 2

Reviewer 1 Report

Thank you very much for the detail on each of the comments made.

Reviewer 2 Report

The manuscript has been revised according to reviewer's comments/suggestions. The paper now is suitable for the publication.